# Fire Resistance of Alkali Activated Geopolymer Foams Produced from Metakaolin and Na_2_O_2_

**DOI:** 10.3390/ma13030535

**Published:** 2020-01-22

**Authors:** Xi Peng, Han Li, Qin Shuai, Liancong Wang

**Affiliations:** 1State Key Laboratory of Fire Science, University of Science and Technology of China, Anhui 230026, China; ccong001@126.com; 2Key Laboratory of Solid Waste Treatment and Resource Recycle, Ministry of Education, Southwest University of Science and Technology, Mianyang 621010, China; lihan1099599@163.com (H.L.); qinshuai_t@163.com (Q.S.)

**Keywords:** alkali activated geopolymer foams, metakaolin, Na_2_O_2_, fire resistance, ceramics

## Abstract

This work aims to investigate the feasibility that alkali-based geopolymer foams produced from metakaolin and Na_2_O_2_ are applied for fire protection. Dry bulk density, porosity, mechanical strength, thermal conductivity, and fire resistance of the geopolymer foams are discussed as a function of the Na_2_O_2_ amounts. As Na_2_O_2_ content varies from 1% to 4%, dry bulk density, mechanical strength and thermal conductivity of the geopolymer foams approximately exhibit opposite trends with that of the porosity. At the later stage of the 3 h fire-resistance tests, the reverse-side temperatures of all tested samples were always maintained at 220–250 °C. Meanwhile, the amorphous skeleton structures have been converted to smooth ceramics during the high temperature processes, which is the main reason that the geopolymer foams possess a stable porous structure and excellent fire resistance. Therefore, we could conclude that alkali-activated geopolymer foams with extraordinary fire resistance have great potential for fire protection applications.

## 1. Introduction

High-strength concrete used in building constructions is unstable when exposed to fire, due to either explosive spalling or overheating of steel reinforcement bars within the lining [1,2]. Fires in cities, mainly happening in buildings and tunnel linings, can seriously damage the concrete lining or even cause collapse of the buildings or tunnels, which may involve multiple deaths or a large loss of property [3]. Massive materials have been explored for building construction protection against high temperature attack, mainly including organic intumescent coatings and cementitious-based coatings [4]. Although organic intumescent coatings are light, aesthetic, and smooth, they cannot provide sufficient protection over a long time in a fire, and may even join in the combustion reaction and produce toxic gases in narrow and long corridors and tunnels at high temperature [3,5]. On the contrary, cementitious-based coatings that cannot combust when exposed to fire may be a more suitable choice in the building construction protection against fire.

Cementitious-based coatings are durable, wear-resistant, inexpensive, easily adhere to concrete, and could provide protection from moisture and high temperature attack [6,7,8]. Generally, intumescent cementitious-based coatings have limitations, such as low mechanical properties and poor spalling resistance that cannot satisfy the requirements of fire protection of building constructions, while the non-intumescent cementitious-based coatings can provide adequate fire protection if they are thick enough [9,10,11,12]. Nevertheless, cement production has the drawbacks of large equipment investment, high energy consumption, air pollution, CO_2_ emission, and high operating cost [13]. More environmentally friendly materials need to be sought for thick coating preparation.

Alkali-activated geopolymers have become a promising alternative cementitious material to ordinary Portland cement that offers attractive properties for commercial applications, i.e., fast hardening, high and early compressive strength, high thermal stability and chemical resistance, and low CO_2_ emission and energy consumption. Indeed, unlike cement spalling at temperatures higher than 200 °C, geopolymers would not undergo significant damage during the dehydration process. Furthermore, geopolymers will convert into aluminate silicate ceramics at temperatures up to 1000 °C, causing a higher mechanical strength. All these features make alkali-activated geopolymers strong candidates for preparing thick coatings. However, thick geopolymer coatings are too heavy, and their construction and maintenance is also a time-consuming process. The lightweight geopolymer foams with excellent thermal insulation performance have gradually gained people’s attention in use as fire-resistant panels, owing to their porous structure and non-combustible characteristic. The application of geopolymer foam panels instead of the thick geopolymer coatings in fire protection may be an effective solution to address the above flaws.

The production of geopolymer foams typically involves the addition of foaming agents (e.g., hydrogen peroxide (H_2_O_2_), aluminium powders) to a geopolymer slurry [14,15]. Hydrogen peroxide is the most popular foaming agent, which can prepare more homogeneous geopolymer foams with volume ratio from 0.25% to 2.5% (volume of H_2_O_2_/volume of geopolymer slurry) [16]. However, hydrogen peroxide is thermodynamically unstable in basic media and is difficult to preserve over the long-term. Although geopolymer foams can be successfully prepared with the addition of aluminum powder as a blowing agent in concentrations from 0.01 wt.% to 0.05 wt.% [17], the generation of explosive gas (H_2_) can bring potential security risks in the preparation process. Therefore, Na_2_O_2_ is put forward in this work as an alternative foaming agent to prepare geopolymer foams. The mechanical properties, thermal conductivity, and the fire protection properties are also discussed.

## 2. Materials and Methods

### 2.1. Materials

The metakaolin was prepared by calcining kaolinite at 850 °C for 2 h. The chemical composition of metakaolin is presented in Table 1. Na_2_O_2_ (analytical reagent grade, Tianjing DaMao Chemical Industry Co., Ltd, Tianjing, China) and sodium dodecyl benzene sulfonate (SDBS, analytical reagent grade, Tianjing Zhiyuan Chemical Industry Co., Ltd, Tianjing, China) were adopted as a chemical foaming agent and a foaming stabilizer, respectively. The sodium silicate solution was adjusted by a certain amount of sodium hydroxide (analytical reagent grade, Guangzhou Huixin Chemical Industry Co., Ltd, Guangzhou, China) to make a solution with a modulus (i.e., molar mass ratio SiO_2_/Na_2_O) of 1.2, which was used as an alkali activator in the geopolymerization process.

### 2.2. Sample Preparation

The weighed metakaolin, alkali activator and water were premixed for 3 min to give complete homogenization. Afterwards, Na_2_O_2_ was added to the mixture and mixed for another 10s. Finally, the mixture was poured into molds with dimensions of 40 × 40 × 40 mm or 40 × 40 × 160 mm (preparation for flexural strength measurement). The samples were then put in a standard curing box at 25 °C with humidity of 90% and under ambient pressure for 24 h. After being removed from the molds, the samples were subjected to curing at room temperature for an additional 27 days. The design of the experiment is shown in Table 2.

### 2.3. Dry Bulk Density, Porosity and Mechanical Characterization

Dry bulk density was determined by the ratio of the geometrical volume and weight, which was obtained after drying each specimen at 105 °C in the oven for 24 h (i.e., so-called geometrical density). Porosity was measured according to the American standard ASTM C642-13. The compressive strengths and flexural strengths of the 28-day cured samples were measured with a CMT5504 compressive strength testing apparatus (Shanghai Heng Yu Instrument Co., Ltd, Shanghai, China). For each measurement, six specimens were tested and the average values were reported.

### 2.4. Thermal Conductivity

The thermal conductivity of the geopolymer foam specimens with a size of 40 × 40 × 15 mm were measured by a thermal conductivity instrument (TC 3000E, Xia Xi technology, Xi’an, China), which is based on the principle of the hot-wire method. In order to ensure good contact of the sample surface, all samples were polished flat and parallel. A mean result was obtained by taking an average of the results from three specimens.

### 2.5. Fire-resistance Tests

The setup for the fire-resistance tests is shown in Figure 1. The fire-resistance tests were carried out by a simulating cabinet method according to GB 12441-2005. During the fire-resistance test, the samples with a size of 40 × 40 × 20 mm were exposed to a 1100 °C flame (liquefied natural gas), and the reverse-side temperatures of the samples were measured by an infrared thermometer.

### 2.6. Microstructural and Mineralogical Characterization

The geopolymer foams before and after fire-resistance tests were characterized via X-ray powder diffraction (XRD) analysis. The XRD pattern was obtained with an X’Pert PRO (PANalytical B.V., Almelo, Netherlands) multifunctional X-ray diffractometer using Cu Kα radiation generated at 50 mA and 45 kV with a scanning step of 0.03° and scanning rate of 10°·min^−1^ step scanned from 3–80° 2-theta. The microscopic analysis of the specimen was conducted using a Zeiss Ultra 55 scanning electron microscope (Carl Zeiss AG, Jena, Germany).

## 3. Results and Discussions

### 3.1. Dry Bulk Density, Porosity and Mechanical Characterization

The dependence of dry bulk density, porosity, compressive strength and flexural strength versus the mass fraction of Na_2_O_2_ are presented in Figure 2. As the figure depicts, the opposite variation between dry bulk density and porosity versus the mass fraction of Na_2_O_2_ from 1% to 4% could be observed. When the mass fraction of Na_2_O_2_ is up to 5%, it is much more difficult to gain an unbroken geopolymer porous block during the demolding process. Although the large mass fraction of the foaming agent is generally considered to be beneficial to the pore formation in geopolymer gels, generation of excessive air bubbles may make the slurry too thin to hold the bubbles. Therefore, controlling addition of Na_2_O_2_ could generally be helpful to tailor the porosity and dry bulk density in the geopolymer foams.

It is also clear from Figure 2 that both compressive strength and flexural strength decline with the decrease of dry bulk density. As a general trend, the introduction of foaming agents would decrease the mechanical strength of the hardened blocks. Moreover, the size and shape of the voids and the strength of the binding skeleton also have causal role in strength development [17,18]. Thus, better control of the foaming agent amounts and the foaming reaction is helpful in forming more homogenous and smaller pores, and gaining lower dry bulk density and higher strength.

### 3.2. Thermal Conductivity

Generally speaking, porosity is considered as the determinant factor for thermal conductivity of porous materials. As we can see from Figure 3, the porosity of the geopolymer foams increases along with the thermal conductivity decreasing substantially. It could be assumed that the pores in geopolymer foams may have influence on the thermal conductivity. In fact, the air with low thermal conductivity in the inner structure of the geopolymer foams plays an important role in reducing the heat-propagation velocity and decreasing the thermal conductivity. However, the decline in thermal conductivity of geopolymer foams suddenly stops when the amount of added Na_2_O_2_ is up to 4% mass fraction. Meanwhile, a larger amount of open and connected pores appears on the surface of porous samples, which may be associated with the sharp change in thermal conductivity [18].

### 3.3. Fire-Resistance Tests

A selection of 20-mm-thick geopolymer foam samples were investigated in this test. Figure 4 shows the results from 3 h fire-resistance tests of the geopolymer foams with various mass fractions of Na_2_O_2_ at a curing time of 28 days. As can be seen from Figure 4, the variation in the reverse-side temperatures of the geopolymer foams can be divided into two processes: (1) the reverse-side temperatures dramatically rise at the initial stage, (2) then the temperatures fluctuate in a very narrow range. At the initial stage, dehydration of the geopolymer gels occurs and causes an irreversible microcrack in the internal structure, resulting in the sharp increases in the reverse-side temperature. Unexpectedly, the skeleton structure of the geopolymer foams is still stable under the later continuous high temperature exposure. Moreover, the stable skeleton structure under high temperatures is beneficial to the heat insulation performance of geopolymer foams. That is why the reverse-side temperatures of geopolymer foams stop rising and always maintain at about 220–250 °C during the later period. As for the 4% N, plenty of the open and connected larger pores in the geopolymer foams make the skeleton structure too thin to maintain the integrity under a continuous high temperature exposure, inducing a quick heat transfer and easily achieving a relative higher temperature in a very short time. Therefore, suitable porosity of the geopolymer foams is one of the vital factors for fire resistance, but the influence of pore characteristics on fire resistance also cannot be ignored.

### 3.4. Microscopic Structure and Mineralogical Characterization

As we can see from Figure 5b, the surface of the porous specimen changes obviously from a rough and amorphous gel to a smooth and crystalline structure after the fire-resistance test. Meanwhile, shrinkage of the skeleton structure occurs and leads to a distinct decrease in pore size, owing to the dehydration and high-temperature sintering. Moreover, nepheline has been detected through X-ray powder diffraction analysis (Figure 5), indicating the ceramization of the skeleton materials in the geopolymer foams after a high temperature exposure. The conversion of the amorphous skeleton materials to a ceramic phase is helpful in enhancing the mechanical strength and porous-structure stability of the geopolymer foams under high temperatures. The robust structure stability of the porous matrices under high temperatures is the main reason that the reverse-side temperatures of geopolymer foams could be maintained at about 220–250 °C during the second stage of the fire-resistance test.

## 4. Conclusions

In this work, alkali-activated geopolymer foams, fabricated with metakaolin and Na_2_O_2_, were investigated as a fire-resistant material for fire protection. As the Na_2_O_2_ content varies from 1% to 4%, dry bulk density of the geopolymer foams decreases gently along with the increase in porosity. Meanwhile, both compressive strength and flexural strength are in correlation with the dry bulk density and porosity. Better control of the foaming agent amounts and the foaming reaction is key in forming more homogenous and smaller pores in geopolymer foams, which is beneficial in gaining lower thermal conductivity and higher strength.

At the later stage of the 3 h fire-resistance tests, the reverse-side temperatures of all tested samples were always maintained at 220–250 °C. Meanwhile, ceramization of the amorphous skeleton materials occurs under high temperature conditions, which may be the main reason that the geopolymer foams exhibit excellent fire resistance. Therefore, the prepared alkali-activated geopolymer foams are promising in the fields of fire-proofing, flame retarding, and so forth.

## Figures and Tables

**Figure 1 materials-13-00535-f001:**
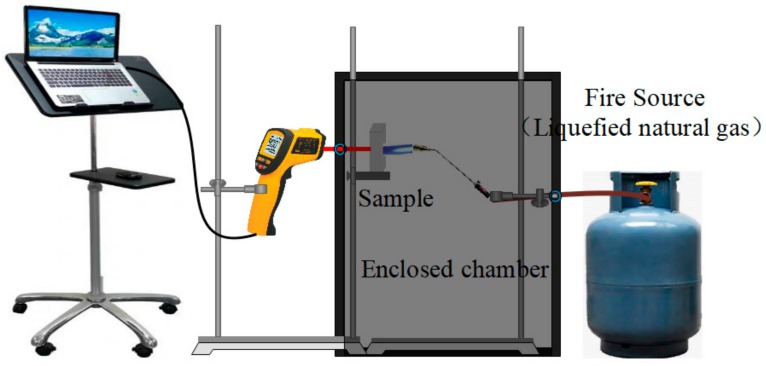
The setup for determining the fire-resistance performance of the sample.

**Figure 2 materials-13-00535-f002:**
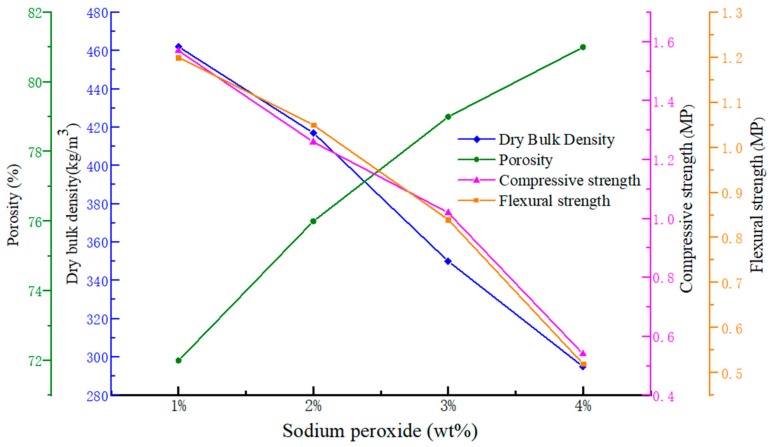
The trend of dry bulk density, porosity, compressive strength and flexural strength versus the mass fraction of Na_2_O_2._

**Figure 3 materials-13-00535-f003:**
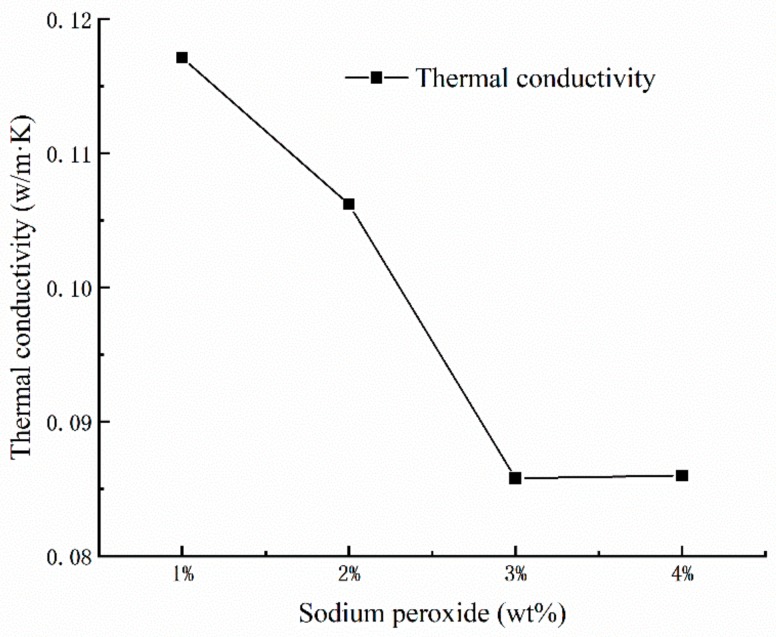
Dependence of thermal conductivity versus the mass fraction of Na_2_O_2_.

**Figure 4 materials-13-00535-f004:**
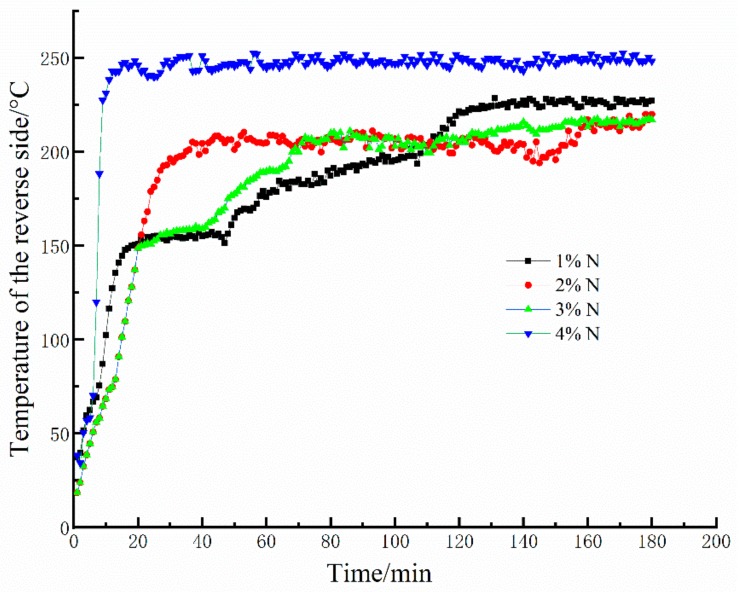
Fire-resistance test results of geopolymer foams.

**Figure 5 materials-13-00535-f005:**
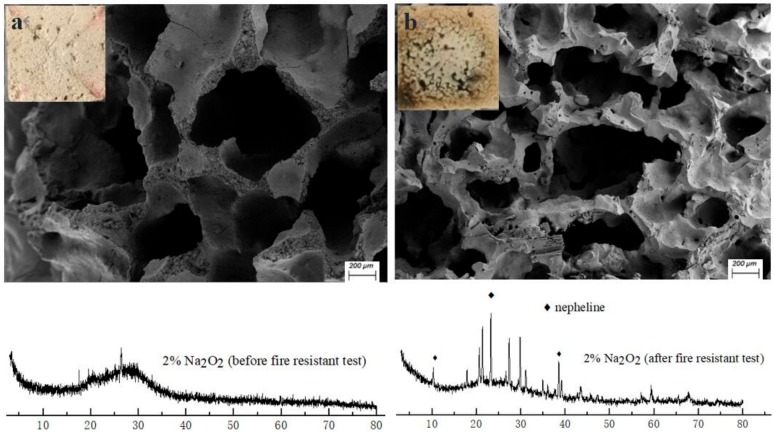
XRD patterns and images of 2% N fraction geopolymer foams: (**a**) before fire-resistance test; (**b**) after fire-resistance test.

**Table 1 materials-13-00535-t001:** Chemical composition of the metakaolin (wt.%).

Component	SiO_2_	Al_2_O_3_	Na_2_O	Fe_2_O_3_	CaO	K_2_O	Others
	54.25	43.92	0.14	0.39	0.13	0.41	0.76

**Table 2 materials-13-00535-t002:** Compositions of the mixtures to prepare geopolymer foams.

Sample Name	Foaming Stabilizer/Na_2_O_2_ (g)	Alkali Activator (g)	Metakaolin (g)	Foaming Stabilizer/SDBS (g)	Distilled Water (mL)
1% N	1.625	50	75	0.3	37.5
2% N	3.250	50	75	0.3	37.5
3% N	4.875	50	75	0.3	37.5
4% N	6.500	50	75	0.3	37.5

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
