# Peer review of "Fire Resistance of Alkali Activated Geopolymer Foams Produced from Metakaolin and Na_2_O_2"

_materials, 2020, doi:10.3390/ma13030535_

Round 1

Reviewer 1 Report

In order to test the fire resistance of a material, the standard fire scenarios should be followed instead of a typical fire of 1100C. In addition the developed material should be applied on a concrete specimen to investigate the behaviour of the material due to this bonding.

In addition i do not know from economical side if the Na2O2 is an agent to be industrially promoted in the material development market.

Author Response

Dear reviewer:

Thank you very much for your good advices. We have revised the manuscript, and would like to re-submit it for your consideration. We have addressed the comments raised by the you, and the amendments are highlighted in red in the revised manuscript. Point by point responses are listed below this letter.

Reviewer 2 Report

The study presents the results from sodium peroxide implementation as a foaming reagent in the production of geopolymers with fire resistant properties. In my opinion the study is interesting for the scientific community. The manuscript is well prepared but needs some efforts to be improved. The detailed comments are presented in the attached file.

Author Response

Dear reviewer:

Thank you very much for your good advices. We have revised the manuscript, and would like to re-submit it for your consideration. We have addressed the comments raised by the you, and the amendments are highlighted in red in the revised manuscript. Point by point responses are listed below this letter.

With best wishes,
Yours sincerely,

Xi Peng

line 25 – 26 The sentence is not clear and should be rewritten to clarify. I recommend replacing the phrase “city constructions” by “building constructions” (throughout whole the text). “Vulnerable” seems not appropriate word for materials attacked by fire or subjected to fire, please consider replacing with more appropriate

Response: We have replaced the phrase “city constructions” by “building constructions” throughout whole the text. “Vulnerable” has been changed to “unstable” (line 25-40).

line 41: consider to replace “the defects” by “drawbacks”

Response: “the defects” has been changed to “drawbacks” (line 42).

line 59 “is not perfect” seems not appropriate to describe the drawbacks of the reagent. Please, rewrite

Response: We have rewritten the sentence according to your valuable advice (line 60-64).

line 61: “…r foams except for generating explosive gas (hydrogen)” is not quite clear. Please, rewrite to clarify

Response: We have rewritten the sentence according to your valuable advice (line 64-66).

line 76: calcining – is it the right notation of the process

Response: Yes, “calcining” is correct here. In fact, calcination of kaolinite is the traditional method to prepare metakaolin.

line 71: a modulus – what do you mean?

Response: Modulus in this article means the molar mass ratio of SiO2/Na2O of water glass (Sodium silicate solution).

line 72: …..an alkali activator in the geopolymerisation process

Response: We have revised the sentence to “……, which was used as an alkali activator in the geopolymerisation process” (line 84-85).

Table 1: remove metakaolin from the second raw

Response:  The metakaolin in the second raw has been removed (line 86).

line 76 with respect to the weight of the slurry – it is not clear. Please, clarify and present details. The experiment should be described in a way which allow everyone to repeat

Response: We have deleted the “with respect to the weight of the slurry”. And the details of experiment parameters were present in the table 2 (line 89,95) .

Table 2 – data in column 2, please present the weight of the added Na2O2 with appropriate significant numbers depending on the precision of the used balance; What “Num.” means?

Response: The added Na2O2 with appropriate significant numbers depending on the precision of the used balance has been presented. “Num.” is short for number.

line98 – please, replace “A mean results was obtained …..)

Response: We have revised the sentence according your good advice (line 107-108).

line 107 – unclean. Please, rewrite

Response: We have rewritten this sentence according your advice (line 122-123).

line 114 – please consider replacing “trend” by “dependence, fuction or other appropriate word”

Response: We have replaced the “trend” to “dependence”.

line 115: are presented on Fig. 2. Remove “tendencies”

Response: We have replaced the “given in Fig.2” to “presented on Fig. 2” (line 131). Meantime, the “tendencies” has been removed (line 132).

line 124 – I don’t think that the presented dependence is “stepwise”. The curve is not smooth, but the steps are not well developed.

Response:  We have removed the “stepwise” (line 140).

line 126: casual ? What do you mean here?

Response: “causal”is adopted just to enhance the causal relationship in “size, shape of the voids and the strength of binding skeleton” and “strength development”.

line 133 generally

Response: We have replaced the “general” to “generally” (line 149).

line 134: “cline chat” – what do you mean?

Response: We have rewritten this sentence (line 150).

line 136: “special effect” What do you mean? May be influence. Please, rewrite

Response: We have modified this sentence (line 152).

line 141: “negative influence” …Why negative? At 4% the thermal conductivity is still low.

Response: We may haven’t expressed the details clearly. Although the thermal conductivity is still low, the decrease tread in thermal conductivity of the geopolymer foams suddenly stops at 4% with the continuous increase of porosity. We have rewritten the two sentences (line155-158).

Figure 3 Dependence/function of thermal conductivity on the mass fraction of Na2O2.

Response: We have modified this sentence according to your good advice (line 161).

line 144 Fig. 4 shows the results from 3h resistant test of geopolymer foams….

Response: We have modified this sentence according to your good advice (line 60-62).

line 148: remove “later period”

Response: “during the later period” has been removed according to your good advice (line 165).

line 149:..microcrack in internal structure

Response: We have modified this sentence according to your good advice (line 166).

line 150: temperature

Response: We have modified this sentence according to your good advice (line 167).

line 151-153 – please rewrite using shorter sentences. It is quite difficult to follow the idea here.

Response: We have rewritten this section according to your good advices. (line 170-175).

Figure 5 : the magnitude is presented only on Fig. 5B. Is it the same on the Figure 5A?

Response: Yes, the magnitude is the same on the Figure 5A. We have revised the Figure 5A according to your good advice (line 197).

lines 166-167: unclear. Please rewrite to clarify

Response: We have rewritten this section according to your good advices (line 189-193).

line 198: Please, follow the same format of references

Response: We have modified the reference according to your good advice (line 229).

Reviewer 3 Report

The article would be of greater value if the review of the literature presents an overview of foaming agents used by other authors with their percentages The fire-resistance test method used is not entirely correct. I have some doubts about the accuracy of the tests carried out. Temeprature measurement and testing should take place in a closed chamber that there is no heat transfer to the environment. Please check the methodology for so-called tests mini jet-fire It should be written clearly in the article that the fire tests carried out were not implemented on the basis of standards and tests that may be subject to some errors due to a rapid heat dissipation An interesting solution would also be publishing the results of photos taken with the infrared camera In the reference list, the titles of the journal should be corrected (font, abbreviations)

Author Response

Dear reviewer:

Thank you very much for your good advices. We have revised the manuscript, and would like to re-submit it for your consideration. We have addressed the comments raised by the you, and the amendments are highlighted in red in the revised manuscript. Point by point responses are listed below this letter.

With best wishes,

Yours sincerely,

Xi Peng

Point 1: The article would be of greater value if the review of the literature presents an overview of foaming agents used by other authors with their percentages. The fire-resistance test method used is not entirely correct. I have some doubts about the accuracy of the tests carried out. Temperature measurement and testing should take place in a closed chamber that there is no heat transfer to the environment. Please check the methodology for so-called tests mini jet-fire It should be written clearly in the article that the fire tests carried out were not implemented on the basis of standards and tests that may be subject to some errors due to a rapid heat dissipation. An interesting solution would also be publishing the results of photos taken with the infrared camera.  In the reference list, the titles of the journal should be corrected (font, abbreviations)

Response 1: Please provide your response for Point 1. (in red)

Thanks very much for your kindness!  We have added other authors’ progress in the end paragraph of the Introduction according to your valuable advices. As for the fire resistance test method, we completely agree with your opinions. In this work, the fire resistance tests were carried out by simulating cabinet method according to GB 12441-2005. Meanwhile, we have realised the influence about heat dissipation and external environmental disturbance for the fire resistance tests. So a enclosed chamber has been adopted to deal with the above problems during the testing process. We will continue to improve the methods and equipment of fire resistance tests according to your advices in the next research programs. We also have finished modified section 2.5. Finally, the errors in the reference list have been corrected as well (line 229).
